# MSLC: Monte Carlo Tree Search Sampling Guided Local Autoregressive Construction for Large-Scale Traveling Salesman Problem

## Abstract

Neural solvers have achieved significant results in solving small-scale Traveling Salesman Problems (TSP), but they are inefficient when handling large instances. Based on the optimal substructure property of the TSP, the solving process can be divided into global selection for the perspective of the whole route and local fine-tuning for the perspective of the sub-route. The autoregressive model-based Local Construction approach fails to explore the global action space well, and the non-regressive model-guided MCTS approach focuses on exploring the global action space, therefore there is still a lot of room for optimisation locally. In order to achieve good results in both global selection and local fine-tuning, we propose the MSLC (**M**onte Carlo Tree Search **S**ampling Guided **L**local Autoregressive **C**onstruction) framework, which innovatively integrates the prediction sampling module into MCTS (Monte Carlo Tree Search) to achieve efficient fusion with local autoregressive construction. Taking advantage of the scalability of MCTS and the accuracy of the autoregressive model, the global selection and local fine-tuning steps are taken into account, and the Sampling module is used to balance the speed of MCTS and local autoregressive construction, optimizing the effect without losing time, greatly improving efficiency. MCTS can be guided by non-autoregressive models, and this framework provides a new combination method for autoregressive and non-autoregressive models. Experimental results demonstrate that MSLC effectively balances time and solution quality, outperforming state-of-the-art neural solvers. The performance gap of MSLC is reduced by at least 29.4% (resp. 34.7% or 28.5%) on TSP-500 (resp. TSP-1000 or TSP-10000), compared to SOTA neural methods.

## 1 Introduction

The Travelling Salesman Problem (TSP) is recognized as a classic combinatorial optimization problem with wide-ranging applications in such as logistics (Madani et al., 2021), chip manufacturing (Kumar & Luo, 2003), or supply chain (Rao, 2017). The task involves determining the shortest possible route that visits a set of cities exactly once before returning to the starting point. Traditional solvers, such as Concorde (Applegate et al., 2009) and LKH-3 (Helsgaun, 2017; Taillard & Helsgaun, 2019), are based on heuristics derived from mathematical methods, requiring extensive computational effort and expert domain knowledge. While high effectiveness has been demonstrated by these solvers for smaller instances, limitations in scalability to larger, real-world datasets have been observed due to the reliance on complex, hand-crafted rules and significant computational demands.

In recent years, neural solvers have become increasingly popular for solving the TSP problem. Compared to traditional solvers, neural solvers are characterized by their ability to learn quickly and iteratively. Based on the optimal substructure property of the TSP (Papadimitriou, 1977), the route $\boldsymbol{\pi} = (\pi_1, \pi_2, \ldots, \pi_N)$ can be improved by adjusting the sub-route $(\pi_i, \ldots, \pi_j)$. In other words, improving the local structure leads to a better global solution. So the process of solving large-scale TSP problems can be divided into two parts: global selection and local fine-tuning. Global selection focuses on optimisation from the perspective of the whole route, local fine-tuning focuses on opti-

misation from the perspective of the sub-route. Good global selection determines the breadth of the result, and good local fine-tuning determines the depth of the result; both are equally important.

Existing neural solvers can be classified into autoregressive construction heuristics solvers and non-autoregressive construction heuristics solvers. Autoregressive Construction Heuristics Solvers face the challenge of high time and space complexity in large-scale TSPs due to the sequential generation scheme of autoregressive models and the quadratic complexity of self-attentive mechanisms. To address this challenge, Kim et al. (2021); Pan et al. (2023); Ye et al. (2024) simply selects global routes and focuses on optimising local sub-routes using Divide And Conquer, failing to fully explore the global path space. Non-autoregressive Construction Heuristics Solvers to solve this scalability issue by assuming conditional independence among variables in TSP, but this assumption limits the ability to capture the multimodal nature (Gu et al., 2017; Khalil et al., 2017) of high-quality solution distributions. Fu et al. (2021); Qiu et al. (2022); Min et al. (2023); Sun & Yang (2023) use Monte-Carlo Tree Search (MCTS) to further improve the expressive power of the non-autoregressive scheme. They focus on searching global routes according to these heatmaps, but leaving significant optimization space for local sub-routes. In summary, existing neural solvers focus on either global or local levels. In order to do well at both, we introduce MSLC.

Both global selection and local fine-tuning are important, but since good global selection depends on the scalability of MCTS, a large number of actions need to be simply searched to obtain a better solution, while good local fine-tuning depends on the accuracy of the sequential generation of the autoregressive model, and a large number of computationally intensive constructions are performed to obtain a better solution. If one wants to achieve good results in both global selection and local fine-tuning by combining MCTS and autoregressive models, the combination of the two will lead to slower MCTS exploration and make the global selection step less effective due to the speed mismatch between MCTS and autoregressive models.

To address this challenge, we introduce the MSLC framework, which effectively fuses MCTS for global selection and local autoregressive construction for local fine-tuning. The sampling module estimates the impact of subsequent tuning and discards part of the initial routes generated by global selection, terminating the process early, thus balancing the speed of global selection and local fine-tuning, optimising the results without loss of time, and improving the efficiency significantly. Specifically, the MSLC framework combines MCTS for global selection with local autoregressive construction for local fine-tuning, and evaluates the initial routes generated by MCTS through 2-opt, because 2-opt can quickly and simply evaluate the optimization space of the initial route. If the adjusted route is far away from the current optimal route, the MCTS search process of some initial routes is terminated early, saving time. Notably, local autoregressive construction allows further route optimisation based on MCTS, while sampling raises the threshold for the initial routes generated by MCTS, thus achieving mutual enhancement. The ablation study shows that the proposed MSLC framework significantly improves the performance and effectively enhances the results without sacrificing time.

**Contributions:**

- This paper proposes the MSLC framework, which effectively combines the scalability of MCTS and the precision of Local autoregressive construction by incorporating a Sampling module into MCTS. This balances the speed of global selection and local fine-tuning, optimising the results without loss of time, and improving the efficiency significantly. The framework offers a novel perspective for problems with optimal substructure, enabling early filtering of global selections by estimating local fine-tuning effectiveness during the global selection process.

- Our method provides an effective way to combine autoregressive and non-autoregressive models. MCTS can be guided by non-autoregressive models, and Local autoregressive construction is based on autoregressive models, laying the foundation for future research on large-scale TSP problems.

- Experiments on TSP-500/1000/10000 demonstrate that the performance gap of MSLC is reduced by at least 29.4% (resp. 34.7% or 28.5%) on TSP-500 (resp. TSP-1000 or TSP-10000) compared to state-of-the-art neural methods.

## 2 RELATED WORK

### 2.1 AUTOREGRESSIVE CONSTRUCTION HEURISTICS SOLVERS

After achieving great success in the field of NLP, autoregressive models have gradually been applied to combinatorial optimization. However, due to the sequential generation scheme of autoregressive models and the quadratic complexity of the self-attention mechanism, these models face significant challenges in both time and space complexity when applied to large-scale TSP problems. Given the optimal substructure property of the TSP, the divide-and-conquer approach has been adopted for solving large-scale TSP. LCP (Kim et al., 2021) was the first to propose a decomposition and reconstruction method, using seeders (autoregressive models) to construct initial routes, followed by Local Construction. However, due to the speed limitations of seeders, it is difficult to scale this approach to large problem sizes. GLOP (Ye et al., 2024) replaced the seeders in LCP with random sampling, and then applied autoregressive models for decomposition and reconstruction, achieving a reasonable solution in a shorter time. H-TSP (Pan et al., 2023) introduced a hierarchical policy for interleaving route selection with Local Construction. Select and Optimize (Cheng et al., 2023) proposed a destruction and repair technique to avoid getting trapped in local optima from a global perspective.

### 2.2 NON-AUTOREGRESSIVE CONSTRUCTION HEURISTICS SOLVERS

Non-autoregressive neural solvers address large-scale TSP problems by assuming conditional independence between variables. However, this assumption often leads to suboptimal local solutions, making additional exploration necessary to enhance the expressiveness of non-autoregressive methods. Monte Carlo Tree Search (MCTS) (Coulom, 2006; Browne et al., 2012; Silver et al., 2016; 2017) is a versatile, adaptive algorithm applicable across various domains. It excels at fully exploring the action space under the guidance of non-autoregressive models, offering significant scalability. ATT-GCN (Fu et al., 2021) combines MCTS with Graph Convolutional Networks (Joshi et al., 2019) by training GCN through supervised learning on small-scale TSP instances. It then generalizes to larger TSPs by generating sub-heatmaps, which are merged into a global heatmap. MCTS, guided by the heatmap, effectively handles large-scale TSP problems. DIMES (Qiu et al., 2022) introduced a compact continuous space to parametrize the underlying distribution of candidate solutions and proposed a meta-learning framework for combinatorial optimization instances. This framework generates an approximate proxy distribution close to the true solution distribution for TSP, though it takes much longer to compute solutions compared to the method by Fu et al. (2021). UTSP (Min et al., 2023) employs an unsupervised learning framework using graph neural networks to generate heatmaps. Its objective function consists of two parts: one encourages the identification of the shortest path, and the other ensures that the solution forms a Hamiltonian cycle covering all nodes. DIFUSCO (Sun & Yang, 2023) leverages the strengths of diffusion models to generate heatmaps for high-quality solutions in combinatorial optimization. DIFUSCO enhances the generation process by proposing TSP problems in the discrete $\{0, 1\}$-vector space and applying denoising diffusion techniques with Gaussian and Bernoulli noise. SoftDist (Xia et al., 2024) is a heatmap generation method that improves the MCTS process for solving large-scale TSPs. It evaluates the effectiveness of heatmaps in guiding MCTS by focusing on the probability distribution of edges belonging to the optimal solution. Compared to various complex machine learning methods, SoftDist demonstrates superior performance by emphasizing the generation of theoretically sound and practical heatmaps, thereby improving the efficiency of strategies for solving combinatorial problems.

Inspired by the strong performance of Autoregressive Construction Heuristics Solvers in local fine-tuning and the effectiveness of Non-autoregressive Construction Heuristics Solvers in global selection, we propose the MSLC framework. MSLC integrates MCTS, guided by non-autoregressive models, with local autoregressive construction based on autoregressive models through the Sampling module. This combination balances speed of global selection and local fine-tuning, significantly improving overall efficiency.

## 3 FORMULATION OF LARGE-SCALE TRAVELING SALESMAN PROBLEM

We focus on the classical two-dimensional Euclidean distance Traveling Salesman Problem (TSP) by defining the TSP problem space as $n$ vertices in a two-dimensional space, denoted by $s = \{x_i\}_{i=1}^N$, where $x_i \in [0,1]^2$. The objective is to find a permutation $\boldsymbol{\pi} = (\pi_1, \pi_2, \ldots, \pi_N)$ that forms a path visiting each vertex exactly once and returning to the starting point. The goal is to minimize the total path length $L(\boldsymbol{\pi})$, computed as follows:

$$L(\pi) = cost(\pi_N, \pi_1) + \sum_{i=1}^{N-1} cost(\pi_i, \pi_{i+1}). \tag{1}$$

## 4 METHOD

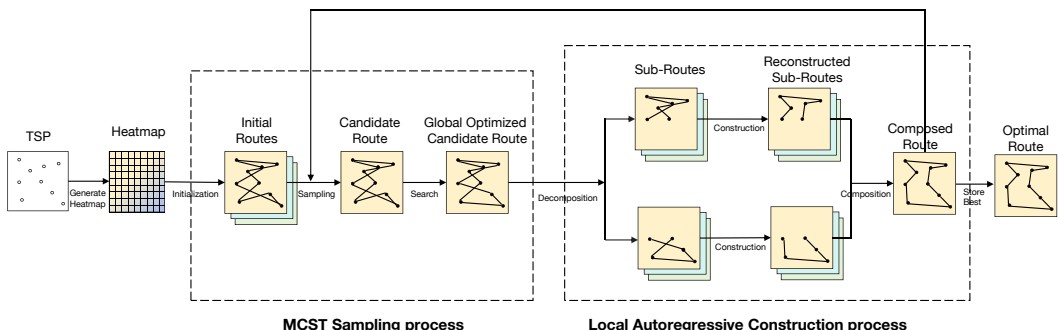

Figure 1: Pipeline.

This section describes a novel hierarchical fusion framework called MCTS Sampling Guide Local Autoregressive Construction, which balances global selection with local fine-tuning through the Sampling module (see Figure 1 for detail). During the global selection process, the MCTS Sampling strategy generates candidate routes and performs evaluation sampling. In the local fine-tuning phase, the Local Autoregressive Construction strategy reconstructs the sub-routes of the sampled candidate routes to minimize the overall candidate routes. The final route is selected as the optimal route among the candidate routes.

### 4.1 MCST SAMPLING PROCESS

In the global selection, the MCTS Sampling strategy generates an initial route based on the heatmap. It uses 2-opt to estimate the initial route and samples the initial route based on the current optimal candidate route. Routes that deviate significantly from the current optimal route are filtered out. The sampled initial routes are then searched and saved as candidate routes.

**Heatmap.**  To explore the global routing space effectively, we generate a heatmap to guide the exploration process. The heatmap is an $N \times N$ symmetric matrix, where $W_{i,j}$ represents the correlation between vertex $i$ and vertex $j$. Higher values indicate a greater likelihood that vertices $i$ and $j$ will be adjacent in the solution. Two methods are provided for generating the heatmap, depending on the trade-off between speed and quality. The first method, based on diffusion, follows Sun & Yang (2023). We apply Bernoulli sampling on a trained non-autoregressive diffusion model to generate discrete variables $x$ as the heatmap. This approach produces high-quality heat maps, but at a slower speed. The second method, following Xia et al. (2024), calculates edge scores to form the heat map using the formula below:

$$\Phi_{i,j} = \frac{e^{-d_{i,j}/\tau}}{\sum_{k \neq i} e^{-d_{i,k}/\tau}}, \tag{2}$$

Compared to the first method, this method generates the heatmap faster but at the cost of some quality. Global selection is based on the heatmap, and a high-quality heat map can more efficiently guide us to find a high-quality initial route and direct the MCTS process.

**Initialization.** This step generates the initial routes based on the heatmap. The global action space is very large, and the initial routing plays a decisive role. We define two $n \times n$ symmetric matrices: the weight matrix $W$ (whose element $W_{ij}$ is initialized to $100 \times H_{ij}$, controlling the probability of selecting vertex $j$ after vertex $i$) and the access matrix $Q$ (whose element $Q_{ij}$ is initialized to 0, recording the number of times that edge $(i, j)$ has been selected during the simulation). In addition, the variable $M$, initialized to 0, is used to record the total number of operations simulated. The weight $Z_{i,j}$ of each edge during the initial route construction is calculated as follows:

$$Z_{i,j} = \frac{W_{i,j}}{\Omega_i} + \alpha \sqrt{\frac{\ln(M + 1)}{Q_{i,j} + 1}}, \tag{3}$$

where $\Omega_i$, the average weight of edges connected to vertex $i$, is defined as $\Omega_i = \frac{\sum_{j \neq i} W_{i,j}}{\sum_{j \neq i} 1}$. Here, $\alpha$ balances exploitation and exploration, and $M$ is the total number of actions sampled so far. The formula for the initial route construction probability is given by:

$$p(\pi) = p(\pi_1) \prod_{i=2}^{n} p(\pi_i | \pi_{i-1}), \tag{4}$$

where $\pi_1$ is chosen at random, and $p(\pi_i | \pi_{i-1})$ is the conditional probability of choosing the next vertex, calculated by the edge potential: $p(\pi_i | \pi_{i-1}) = \frac{Z_{\pi_{i-1}, \pi_i}}{\sum_{l \in \mathbb{X}_{\pi_{i-1}}} Z_{\pi_{i-1}, l}}$ with $\mathbb{X}_{\pi_{i-1}}$ includes candidate vertices connected to $\pi_{i-1}$, selected based on their edge potential value.

**Sampling.** In this step, the initial routes are sampled, discarding those that are unlikely to become optimal after optimization. The sampled routes are then saved as candidate routes. Specifically, the probability of an initial route being saved as a candidate route is as follows:

$$p(\pi) = \begin{cases} 1, & \text{if } L(\pi) - I_{\text{2-opt}} - G > L(\pi_{\text{best}}), \\ 0, & \text{otherwise.} \end{cases} \tag{5}$$

where $I_{2-opt}$ represents the change in length after applying the 2-opt adjustment to the initial route, and $G$ is a parameter used to control the sampling intensity. This step discards most of the suboptimal initial routes, saving time on further adjustments.

**$k$-opt Search.** This step performs a global coarse-grained optimization on the candidate routes based on $k$-opt moves. Each $k$-opt move is represented as a vertex decision sequence $(a_1, b_1, a_2, b_2, \ldots, a_k, b_k, a_{k+1})$, where $a_{k+1} = a_1$. This sequence involves removing $k$ edges $(a_i, b_i)$ and adding $k$ new edges $(b_i, a_{i+1})$, for $1 \leq i \leq k$. After selecting $b_i$, the next vertex $a_{i+1}$ is sampled according to Equation 4. The route $\pi$ is transformed into $\pi_{\text{new}}$, and the metrics $M, Q_{b_i, a_{i+1}}$, and $Q_{a_{i+1}, b_i}$ are updated accordingly.

**Back-propagation.** For each candidate route optimized by $k$-opt search, the matrices $W$ and $Q$, as well as the global action counter $M$, are updated. The matrix $W$ is updated only for actions that lead to improved states, thereby increasing the probability of these actions being selected in future iterations.

## 4.2 LOCAL AUTOREGRESSIVE CONSTRUCTION PROCESS

In the local fine-tuning, the Local Autoregressive Construction strategy will use autoregressive models based on scales of 20/50/100 to iteratively reconstruct candidate routes $T_{20}/T_{50}/T_{100}$ times. For each candidate route, it is decomposed into sub-routes according to the applicable scale of the autoregressive model and the number of reconstruction iterations $T$. Then, each sub-route is reconstructed using AM(Kool et al., 2019), and the better sub-routes before and after reconstruction are retained. Finally, the sub-routes are merged to form a new candidate route.

**Model Traning.** To train the autoregressive model for Construction, we used the AM architecture proposed by Kool et al. (2019) and trained using rollout baseline based reinforcement learning. Also inspired by POMO (Kwon et al., 2020), we exploit the symmetry of the sub-routes. The path from the head to the tail versus the path from the tail to the head, and their average values can define the rollout baseline more accurately. so during training, our algorithm forces the network to set the starting point as the head and the tail in the same batch of data, and the training loss function is defined as follows:

$$\nabla L(\theta|s) = \mathbb{E}_{p_\theta(\pi|s)} \left[ (R(\pi) - b(s)) \nabla \log p_\theta(\pi|s) \right] \tag{6}$$

where the reward function $R(\pi) = -L(\pi)$ and $b(s)$ represents the average reward of the batch data.

**Decomposition.** To enable the autoregressive model to optimize large-scale routes locally, we adopt a divide-and-conquer approach to decompose the large-scale routes. The goal of decomposition is to comprehensively cover sub-routes of size $M$ based on the number of iterations $I$ and the scale $M$ that the autoregressive model can handle. In the first iteration, we randomly select a starting point and divide the route into $N/M$ sub-routes, while the remaining segment of length $N \mod M$ is kept unchanged. In subsequent iterations, we identify $M/I$ points to the right of the starting point from the last iteration and use one of these points as a new starting point to re-decompose the route into sub-routes. This decomposition step aims to maximize the optimization space within a limited number of iterations.

**Construct.** For each decomposed sub-route, we use the autoregressive model (AM) for reconstruction. We simultaneously select both endpoints of the sub-route as the starting points for reconstruction, following the strategy:

$$p_R(\pi_{k+1:k+l}|s) = \prod_{t=1}^{l} p_{\theta R}(\pi_{k+t}|\pi_{k:k+t-1}, \pi_{k+l+1}, s) \tag{7}$$

where $p_{\theta R}$ is parameterized by the autoregressive model trained in the Model Training module.

**Composition.** We compare the initial sub-route with the two reconstructed sub-routes and retain the $L$ shortest sub-routes. The retained sub-routes are then connected to the tail sub-route at their endpoints, and the merged route is saved as a candidate route.

## 5 EXPERIMENT

### 5.1 EXPERIMENTAL SETTINGS

**Datasets** To evaluate the efficiency of MSLC, we compare its performance against state-of-the-art (SOTA) methods using the same instances. Specifically, we assess MSLC on uniformly sampled large-scale instances of TSP500, TSP1000, and TSP10000, as utilized in the study by Fu et al. (2021).

**Settings** During the generation of the heat map, we adopted the same parameter settings as Xia et al. (2024). In the MCTS sampling process, for TSP500/1000/10000, we set $G = 1/2/10$; during the local construction process, for TSP500/1000/10000, we set $T_{20} = 2/2/5$, $T_{50} = 5/5/25$, and $T_{100} = 5/5/20$. For the generation of the autoregressive model, we used the same hyperparameters as Kool et al. (2019).

**Evaluation Metrics** We use three metrics to compare the performance of different solutions: average trip length (Length), average relative performance gap (Gap), and total run time (Time). Notably, the total runtime of the heatmap-based solution encompasses both the heatmap generation time and the search time.

**Hardware** MSLC and the baseline methods are executed on a 64-core AMD EPYC 7T83 Processor and an NVIDIA RTX 4090 Graphics Card. We utilize as many threads as possible to prevent the CPU from idling while waiting for GPU computations. Specifically, we employ 128 threads for TSP500 and TSP1000, and 16 threads for TSP10000.

Table 1: Results on large-scale TSP problems. Some methods list two terms for Time, corresponding to heatmap generation and others.

| Method | TSP-500 | | | TSP-1000 | | | TSP-10000 | | |
|---|---|---|---|---|---|---|---|---|---|
| | Length | Gap | Time | Length | Gap | Time | Length | Gap | Time |
| Concorde | 16.55* | — | 37.66M | 23.12* | — | 6.65H | N/A | N/A | N/A |
| LKH-3 | 16.55 | 0.00% | 46.28M | 23.12 | 0.00% | 2.57H | 71.78* | — | 8.8H |
| Gurobi | 16.55 | 0.00% | 45.63H | N/A | N/A | N/A | N/A | N/A | N/A |
| Farthest Insertion | 18.30 | 10.57% | 0s | 25.72 | 11.25% | 0s | 80.59 | 12.29% | 6s |
| AM | 22.64 | 36.84% | 15.64M | 42.80 | 85.15% | 63.97M | 431.58 | 501.27% | 12.63M |
| POMO+EAS-Emb | 19.24 | 16.25% | 12.80H | N/A | N/A | N/A | N/A | N/A | N/A |
| POMO+EAS-Lay | 19.35 | 16.92% | 16.19H | N/A | N/A | N/A | N/A | N/A | N/A |
| POMO+EAS-Tab | 24.54 | 48.22% | 11.61H | 49.56 | 114.36% | 63.45H | N/A | N/A | N/A |
| InViT | N/A | N/A | N/A | 24.65 | 6.62% | 4.80M | 76.14 | 6.08% | 10.30M |
| H-TSP | N/A | N/A | N/A | 24.57 | 6.31% | 0.78M | 77.75 | 7.32% | 0.79M |
| Select and Optimize | 16.94 | 2.40% | 0.25M | 23.76 | 2.80% | 0.42M | 74.29 | 3.51% | 7.61M |
| GLOP | 16.91 | 1.99% | 1.50M | 23.84 | 3.11% | 3.00M | 75.29 | 4.90% | 1.80M |
| UTSP | 16.68 | 0.83% | 3.04M (1.37M+1.67M) | 23.39 | 1.18% | 6.69M (3.35M+3.34M) | N/A | N/A | N/A |
| ATT-GCN | 16.82 | 1.64% | 2.19M (0.52M+1.67M) | 23.67 | 2.37% | 4.07M (0.73M+3.34M) | 74.50 | 3.80% | 20.94M (4.16M+16.78M) |
| DIMES | 16.84 | 1.77% | 2.64M (0.97M+1.67M) | 23.68 | 2.44% | 5.42M (2.08M+3.34M) | 74.10 | 3.23% | 21.43M (4.65M+16.78M) |
| SOFTDIST | 16.78 | 1.44% | 1.67M (0.00M+1.67M) | 23.63 | 2.20% | 3.34M (0.00M+3.34M) | 74.03 | 3.13% | 16.78M (0.00M+16.78M) |
| DIFUSCO | 16.63 | 0.51% | 5.28M (3.61M + 1.67M) | 23.39 | 1.18% | 15.20M (11.86M+3.34M) | 73.76 | 2.77% | 45.29M (28.51M+16.78M) |
| OURS(SOFTDIST) | 16.71 | 0.96% | **1.67M (0.00M+1.67M)** | 23.51 | 1.68% | **3.34M (0.00M+3.34M)** | 73.46 | 2.32% | **16.78M (0.00M+16.78M)** |
| OURS(DIFUSCO) | **16.61** | **0.36%** | 5.28M (3.61M + 1.67M) | **23.30** | **0.77%** | 15.20M (11.86M+3.34M) | **73.21** | **1.98%** | 45.29M (28.51M+16.78M) |

## 5.2 BASELINES

For the baselines, we use three types of methods: traditional heuristics, autoregressive construction heuristics, and non-autoregressive construction heuristics. For traditional heuristics, we use LKH (Helsgaun, 2017), Concorde (Applegate et al., 2009) and the commercial solver Gurobi,, which focus on effectiveness, and farthest insertion (Golden et al., 1980), which emphasizes speed. For autoregressive construction heuristics, we use AM (Kool et al., 2019), POMO (Kwon et al., 2020), and InVit (Fang et al., 2024), which are based on sequential generation. Additionally, we use GLOP (Ye et al., 2024), H-TSP (Pan et al., 2023), and Select and Optimize (Cheng et al., 2023), which are based on divide-and-conquer strategies. For non-autoregressive construction heuristics, we employ ATTGCN (Fu et al., 2021), DIMENS (Qiu et al., 2022), SOFTDIST (Xia et al., 2024), DIFUSCO (Sun & Yang, 2023), and UTSP (Min et al., 2023) to guide MCTS. Our focus is to evaluate the ability of MSLC to explore whether combining autoregressive and non-autoregressive models leads to better performance than using autoregressive or non-autoregressive models alone. Since MSLC uses SOFTDIST and DIFUSCO to bootstrap MCTS, we specifically compare its results with Xia et al. (2024) and Sun & Yang (2023). Additionally, since the local autoregressive construction in MSLC is based on Divide and Conquer, we concentrate on comparing the results with Ye et al. (2024), Pan et al. (2023) and Cheng et al. (2023).

## 5.3 RESULTS

The experimental results are shown in Table 1, we firstly focus on the result comparison between MSLC and autoregressive constructive heuristics, and find that MSLC far exceeds autoregressive constructive heuristics in the test results of TSP500/1000/10000, and the experimental results proved that MSLC, compared to the autoregressive constructive heuristics, has a greatly improved compared to autoregressive construction heuristics. Secondly, we focus on the comparison between MSLC and the non-autoregressive construction heuristic, and find that MSLC also improves substantially in the TSP500/1000/10,000 test results, and the experimental results prove that MSLC has a better balance between global selection and local fine-tuning than the non-autoregressive construction heuristic.

## 5.4 ABLATION STUDIES

In this section, we perform ablation studies on MSLC components. In Table 2, we perform ablation experiments on three important components, MCTS, Sampling, and Local Autoregressive Construction, and show the results for each case. We find that the combination of MCTS and Local Autoregressive Construction does enhance the effect in some cases, while the introduction of the

Sampling module further enhances the effect, suggesting that the Sampling module enables MCTS and Local Autoregressive Construction.

Table 2: Ablation study of MSLC components on TSP ($N = 500/1000/10000$). The optimal gap is measured by comparing it with LKH-3. MCTS is guided by DIFUSCO and Local Autoregressive Construction is based on AM. The best performances are marked in bold.

| Component of the MSLC | | | TSP500 | | TSP1000 | | TSP10000 | |
|---|---|---|---|---|---|---|---|---|
| MCTS | Sampling | Local Autoregressive Construction | cost | gap | cost | gap | cost | gap |
| ✓ | | | 16.63 | 0.51% | 23.39 | 1.18% | 73.76 | 2.77% |
| | | ✓ | 16.91 | 1.99% | 23.84 | 3.11% | 75.29 | 4.90% |
| ✓ | | ✓ | 16.63 | 0.51% | 23.37 | 1.08% | 73.55 | 2.45% |
| ✓ | ✓ | ✓ | **16.61** | **0.36%** | **23.30** | **0.77%** | **73.21** | **1.98%** |

# 6 CONCLUSION AND FUTURE WORK

## 6.1 CONCLUSION

In this paper, we propose a novel DRL scheme, i.e., MSLC. based on the nature of the optimal substructure of the TSP problem, the solution process is divided into coarse granularity selection and fine-grained fine-tuning. Based on the scalability of MCTS to explore the coarse granularity selection action space as much as possible, and the accuracy of autoregressive model to optimise the fine granularity fine-tuning action space as much as possible. The scalability of MCTS and the accuracy of autoregressive model are effectively combined by introducing the Sampling module. substantially improve the experimental results without loss of speed, and outperform the current SOTA on large-scale TSP problems.

## 6.2 FUTURE WORK

Future research will focus on two areas. The first area will continue to focus on solving the large-scale traveller problem, and further research will be directed towards the introduction of a more sophisticated Sampling module to combine the MCTS for global selection with the autoregressive model for local construction, enhancing the scalability of the fused MCTS with the accuracy of the autoregressive model. In the second area, other problems with optimal substructure properties will be solved based on MSLC. For any problem with optimal substructure, the solution process can be divided into two steps: global selection and local fine-tuning, exploring the global selection action space as much as possible based on MCTS scalability, constructing local fine-tuning actions as accurately as possible based on autoregressive model accuracy, and solving the problem by using Sampling fusing the scalability of the MCTS with the accuracy of the autoregressive model.

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
