# OpenReview forum: "MSLC: Monte Carlo Tree Search Sampling Guided Local Construction for Solving Large-Scale Traveling Salesman Problem"
_ICLR.cc/2025/Conference — ICLR 2025 Conference Withdrawn Submission_

### Official Review · Reviewer_ggjh · 2024-10-25

**Soundness:** 3
**Presentation:** 1
**Contribution:** 2
**Rating:** 3
**Confidence:** 4

**Summary:**

This manuscript proposes a two-stage neural network method for solving large-scale Traveling Salesman Problems (TSPs). The experimental results demonstrate the effectiveness of the proposed method compared with baseline methods.

**Strengths:**

S1: This manuscript utilizes baseline models that have been recently proposed.
S2: This manuscript shows the significance of using divide-and-conquer strategies to solve large-scale TSPs

**Weaknesses:**

1: The writing is poor. Lots of sentences and words used are ambiguous and confusing and lack logical coherence. For example, in the introduction: what do you mean by *neural solvers are characterized by their ability to learn quickly and iteratively* in the second paragraph; In the third paragraph, what is the *multimodal nature*. Also, the whole fourth paragraph is confusing.

2: This manuscript lacks innovation and appears to be a mere combination of the prior studies [1, 2].

3: The amount of work in the experimental section of this manuscript appears insufficient, particularly given that the main body of the manuscript is less than 8 pages in length. The authors may consider conducting further experiments on TSPLIB, as done in prior studies [2]

4: It appears that there may be some issues with the results presented in Table 1. Specifically, it is unclear why the total time of OURS (DIFUSCO) and DIFUSCO are identical.


[1] Yifan Xia, et al. Position: Rethinking post-hoc search-based neural approaches for solving large-scale traveling salesman problems. In ICML, 2024.

[2] Haoran Ye, et al. GLOP: Learning global partition and local construction for solving large-scale routing problems in real-time. In AAAI, 2024.

**Questions:**

Please refer to the weakness section.

---

### Official Review · Reviewer_ns56 · 2024-11-04

**Soundness:** 3
**Presentation:** 3
**Contribution:** 2
**Rating:** 5
**Confidence:** 3

**Summary:**

Motivated by recent advances in (non)autoregressive methods for solving large-scale TSPs, this paper proposes a new method for solving large scale instances MSLC, a method to locally refine subtours in a TSP with autoregressive models by first constructing larger tours via MCTS sampling from a heatmap (either learned or constructed with heuristics). The proposed method is evaluated in large-scale TSPs up to 10,000 nodes and demonstrates SOTA results against several neural baselines.

**Strengths:**

The paper is overall well written and quite clear to follow for someone in neural combinatorial optimization (NCO). The paper considers two relevant variants for heatmap generation, i.e., DIFUSCO and SoftDist, following recent work on heatmap generation. Results are good and the paper has significance in a sub-area of NCO focusing on large-scale TSPs.

**Weaknesses:**

1. I believe the biggest weakness of the paper is the quite limited applicability of the method. While authors do not overclaim the generality of the method to other problems, MSLC is only applied to the Euclidean TSP, which already has several solvers available which are much better than NCO approaches. There are two ways that could make the paper more interesting and strengthen it for the NCO community without asking for more constrained vehicle routing variants:

    1. Testing on the asymmetric TSP

    2. Testing the robustness of the proposed method on some more realistic distributions, such as the TSPLib

2. Originality and novelty: another concern is regarding the limited novelty of the proposed approach. Overall, this paper mostly follows SoftDist, i.e. in the formulation section, and adds a sampling procedure to enable local construction. In terms of local construction, several approaches already studied this. Moreover, some important recent work is missing such as UDC [1], which proposes an end-to-end approach to train NAR and AR solvers and demonstrates promising results. I believe this paper should be cited and compared in Table 1. Moreover, I would suggest adding other references for large-scale TSP  solvers such as BQ-NCO [2] and LEHD [3].

3. No code is provided as supplementary material or via anonymized links hindering reproducibility.


---

[1] Zheng, Zhi, et al. "Udc: A unified neural divide-and-conquer framework for large-scale combinatorial optimization problems." NeurIPS 2024.

[2] Drakulic, Darko, et al. "Bq-nco: Bisimulation quotienting for efficient neural combinatorial optimization." NeurIPS 2023.

[3] Luo, Fu, et al. "Neural combinatorial optimization with heavy decoder: Toward large scale generalization." NeurIPS 2023.

**Questions:**

1. Table 2: What would the performance be of just using MCTS and Sampling without local construction?

2. Why not use another method for local construction, say 2-opt or LKH, as GLOP does?

3. The value of $G$ was decided as 1/2/10. How was this chosen? What is the sensitivity of MSLC w.r.t. to this hyperparameter?

4. Table 1: the time for running MSLC is the same as for its backbone heatmap generator. Why? Shouldn’t MSLC take longer since there is an improvement loop of sampling and local construction via autoregressive solvers?

---

### Official Review · Reviewer_gng7 · 2024-11-05

**Soundness:** 2
**Presentation:** 1
**Contribution:** 2
**Rating:** 3
**Confidence:** 5

**Summary:**

The paper introduces the MSLC framework for large-scale Traveling Salesman Problem (TSP) instances, combining Monte Carlo Tree Search (MCTS) for global selection and autoregressive construction for local fine-tuning. It claims that by using a sampling module to integrate these two components, MSLC improves upon state-of-the-art neural solvers both in terms of solution quality and computational efficiency. The authors argue that this combination of autoregressive and non-autoregressive models is a novel approach for large-scale combinatorial optimization, specifically TSP, and presents results indicating improved performance over existing methods.

**Strengths:**

The proposed framework has a solid basis in combining MCTS’s scalability with the autoregressive model's accuracy. The concept of a sampling module to align MCTS exploration speed with autoregressive local adjustments shows some innovation in tackling speed and quality trade-offs, which can be a challenging aspect in combinatorial optimization tasks. The authors also achieve competitive experimental results, showing improvements over recent neural methods, suggesting that the MSLC framework may be beneficial for applications in other large-scale optimization scenarios with optimal substructure properties.

**Weaknesses:**

While the paper introduces an interesting combination of techniques, its overall structure and clarity need substantial improvement. The introduction is overly focused on specific technical details without providing an accessible overview of the proposed approach, which makes it hard for readers unfamiliar with MCTS or autoregressive models to grasp the main idea. An organizational approach that follows the logical flow presented in Figure 1 would make it clearer.

The clarity of the paper is further compromised by several issues, such as the mention of SoftDist at line 216, which feels out of context and disrupts the logical flow of ideas. There are also minor but cumulative technical inconsistencies, such as using $H_{ij}$ without prior definition at line 222. In Equation 5, it may be more effective to switch the positions of $ L(\pi_{\text{best}}) $ and $G$, as this would enhance clarity regarding the role of $G$ in the selection process. Furthermore, some steps in the methodology lack logical consistency with the algorithmic flow—for instance, sampling should logically precede estimation in line 202, as 2-opt isn’t performing estimation but rather sampling.

Beyond structural concerns, the originality of the approach is limited due to its reliance on existing methods. The heatmap generation leverages Sun & Yang (2023), while the use of k-opt in MCTS and section 4.2’s decomposition resemble standard techniques, such as those in LEHD [1]. Additionally, the experimental comparison fails to benchmark against similar methods like LEHD [1], limiting the evaluation of the true effectiveness of MSLC. Consequently, the innovative aspect of MSLC appears diminished, as it is largely an integration of established methods without substantial novel contributions.

[1] Luo, Fu, et al. "Neural combinatorial optimization with heavy decoder: Toward large scale generalization." Advances in Neural Information Processing Systems 36 (2023): 8845-8864.

**Questions:**

Could the authors re-evaluate the introduction to present a higher-level overview of MSLC’s design before delving into technical specifics? This would benefit readers and clarify the framework’s value proposition.

Can the authors provide a comparative analysis against LEHD and other latest methods that able to deal with large size instance? This addition would strengthen the claim of MSLC's effectiveness by situating it more clearly among similar approaches.

---

### Official Review · Reviewer_o78J · 2024-11-08

**Soundness:** 2
**Presentation:** 1
**Contribution:** 1
**Rating:** 3
**Confidence:** 5

**Summary:**

This paper introduces MSLC, a framework that combines heatmap-guided sampling with MCTS and local autoregressive construction to solve large-scale TSP efficiently. By focusing search on high-potential paths, MSLC improves both solution quality and speed, outperforming existing neural solvers on TSP instances up to 10,000 nodes.

**Strengths:**

1. Combining MCTS and local construction to balance exploration and refinement, enhancing solution quality.
2. Heatmap-guided sampling improves search efficiency by focusing on high-potential paths.
3. Outperforms existing solvers on large TSP instances, demonstrating scalability and speed.

**Weaknesses:**

1. The paper exhibits significant deficiencies in writing quality, suggesting a lack of thorough revision. Its structure and language are confusing, with substantial room for improvement in clarity and logical flow. Specific issues include:

1) Poor Writing Quality: The paper’s clarity and organization are severely lacking, making it challenging to follow the methods and results.
Spelling Mistakes: "Llocal" on line 21 should be "Local," and "self-attentive mechanisms" on lines 59-60 should be "self-attention mechanism."
Logical Shortcomings: The Introduction section lacks clear logic and is repetitive. Lines 057-069 introduce MCLS and its purpose, but lines 070-077 rehash previously covered points without adding new information, and lines 078-090 reintroduce MCLS yet again. This unclear logic hinders reader comprehension. The Methods section also lacks a clear, sequential flow, with abrupt transitions, inconsistent terminology, and redundant descriptions, making it difficult to understand how each component functions within the MCLS framework.
2) Lack of Explanation for Notation: The paper uses terms without adequate explanation, which can be misleading. For instance, lines 254-255 introduce $(a_1, b_1, \dots)$ without clarifying what $(a_i, b_i)$ represents, leaving it unclear whether $a_i$ and $b_i$ are nodes in the graph. If they are nodes, this conflicts with the notation on line 166, where nodes are defined as $x_i$. Additionally, parameters like $T_{50}$ are used without prior definition or context, making it difficult for readers to fully understand the methodology. The notation "TSP500/1000/10000" is confusing and potentially misleading, as it resembles multiplication notation.
3) Ambiguous Language and Lack of Clarity: Lines 268-269, "the better sub-routes before and after reconstruction are retained," are unclear in meaning. It’s ambiguous whether this means retaining the best sub-routes among those generated before and after reconstruction, or retaining the sub-routes that show the greatest improvement post-reconstruction.
4) Inconsistent and improper use of abbreviations: Lines 018-019 introduce MCTS without providing the full term until line 023. Besides, "MCTS" appears several times without abbreviation in the main text until line 135. Also, abbreviations should follow the format of full term (abbreviation), e.g., Monte Carlo Tree Search Sampling Guided Local Autoregressive Construction (MSLC). Additionally, "DRL" is directly mentioned on lines 397-398 without prior introduction or explanation, which requires the full term to be included on its first appearance.

2. The novelty of this paper is insufficient: MCTS has long been applied to solving TSP, and most components within the framework are directly borrowed from existing models such as AM, DIFUSCO, and SOFTDIST. The only area of potential improvement is the sampling module, but it offers minimal innovation as it primarily relies on standard 2OPT.

3. Insufficient Ablation Studies: The paper includes minimal ablation studies, and the analysis provided is insufficient. This lack of thorough ablation testing fails to clarify the individual contributions of key components, leaving it unclear how each part of the model affects overall performance.

**Questions:**

Pls refer to the weaknesses.

---

### Note · Authors · 2024-11-15

I have read and agree with the venue's withdrawal policy on behalf of myself and my co-authors.